# Chemical and Sensory Characterization of Xinomavro Red Wine Using Grapes from Protected Designations of Northern Greece

**DOI:** 10.3390/molecules28135016

**Published:** 2023-06-27

**Authors:** Elli Goulioti, David W. Jeffery, Alexandros Kanapitsas, Despina Lola, Georgios Papadopoulos, Andrea Bauer, Yorgos Kotseridis

**Affiliations:** 1Laboratory of Enology and Alcoholic Drinks (LEAD), Agricultural University of Athens, 75 Iera Odos, 11855 Athens, Greece; elligoulioti@gmail.com (E.G.); alex.kanapitsas@gmail.com (A.K.); despinalola@gmail.com (D.L.); 2School of Agriculture, Food and Wine, and Waite Research Institute, The University of Adelaide, PMB 1, Glen Osmond, SA 5064, Australia; 3Laboratory of Plant Breeding and Biometry, Department of Crop Science, Agricultural University of Athens, 75 Iera Odos, 11855 Athens, Greece; gpapadop@aua.gr; 4Department of Food Science and Nutrition, Faculty Life Sciences, Hamburg University of Applied Sciences, Ulmenliet 20, 21033 Hamburg, Germany; andrea.bauer@haw-hamburg.de

**Keywords:** aroma, GC-MS, GC-MS/olfactometry, sensory descriptive analysis, volatile compounds, modified frequency, protected designation of origin

## Abstract

Despite Xinomavro (*Vitis vinifera* L.) being a well-known noble red grape variety of northern Greece, little is known about its ‘‘bouquet’’ typicity. Volatile compounds of Xinomavro wines produced using a common vinification protocol were analyzed by gas chromatography–mass spectrometry and sensory descriptive analysis was carried out with a trained panel. Wines were characterized by the presence of fatty acids, ethyl and acetate esters, and alcohols, with contributions from terpenes and a volatile phenol. The most active aroma compounds were determined to be 3-methylbutyl acetate, β-damascenone, ethyl esters of octanoic and hexanoic acids, and eugenol. Those compounds positively correlated with fruity and spicy odor descriptors, with the wines being mostly characterized by five typical aroma terms: strawberry, berry fruit, spices, tomato, and green bell pepper. Partial least squares regression (PLSR) analysis was used to visualize relationship between the orthonasal sensory attributes and the volatile aroma compounds with calculated OAVs > 1. Key aroma-active volatiles in the wines were identified using GC-MS/olfactometry, providing a list of 40 compounds, among which 13 presented a modified detection frequency > 70%. This study is the first of its kind and provided strong indications regarding the aroma compounds defining the sensory characteristics of Xinomavro wines.

## 1. Introduction

From antiquity to today, knowledge of viticulture and enology in Greece has yielded many red grape *Vitis vinifera* varieties [1] that have attracted international attention due to their unique and distinct characteristics [2,3]. Nowadays, there is an interest in understanding and exploiting regional uniqueness of grape varieties [4]. In particular, native Greek variety Xinomavro (*Vitis vinifera* L.) is the flagship red variety of northern Greece and is cultivated in the winegrowing regions of Amyndeon and Naoussa, producing protected designation of origin (PDO) wines [3]. Xinomavro produces black grapes of high acidity [1], maturing in late September and early October [5]. The subsequent wines are generally light in color [6] but rich in tannins [1], with sensory descriptors linked to astringency, sour taste, and bitter aftertaste [5]. These wines generally present many sensory similarities to Nebbiolo wines [2] or the aroma bouquet of Burgundy reds [5].

The quality of wines is a multidimensional concept for which sensory characteristics have a significant role [4]. This depends frequently on the unique properties of one or more grape varieties, geographical origin, and vintage [7]. Characterizing the unique sensory traits of Greek PDO wines such as Xinomavro is necessary, not only for the certification of wines from protected regions, but also for the creation of novel offerings that are well suited to a varying climate [4]. The aroma character of wine is the result of a long biological, biochemical, and technological sequence, and consists of hundreds of volatile compounds [7] of different origins [8]. A complex matrix such as wine, therefore, contains diverse chemical constituents, thus requiring sensitive analytical techniques to make chemical characterization efforts successful.

Various studies involving the identification of volatile compounds contributing to the unique sensory properties of wines have been published over the past ten years [9,10,11,12]. To assess a wine’s quality parameters, both analytical and sensory methods are frequently used. The most widely used analytical technique for characterizing the aroma profile of wine is gas chromatography with mass spectrometry (GC-MS), which can also be combined with gas chromatography–olfactometry (GC-O) [13]. Wine extracts are analyzed by GC-O to describe the odor-active zones and highlight important odorants, whereas GC-MS is applied for analyte identification and quantification [14]. The results of sensory assessment in the form of descriptive analysis (DA) can then be related with the volatile compound profiles [14]. Notably, GC-O is a hybrid instrumental–human olfactory analysis technique that differs from classical sensory analysis because odorants can have a synergistic effect perceived during sensory analysis that would not be evident by GC-O because the components are separated [9]. Combining the two sensory approaches with quantitative GC-MS analysis of the volatiles that provide favorable sensory properties [15] is an effective approach that could also produce significant advances in understanding the aroma profiles of Xinomavro varietal wines.

Xinomavro wines have recently represented Greece on the international stage [3], but the particular volatile compounds that contribute to the typical aromas of these wines have not been reported so far. For significant progress in managing and enhancing wine quality, knowledge of the compounds responsible for determining aroma characteristics in wines is extremely desirable [16]. Establishing sensory–instrumental analysis relationships is a key goal of research that attempts to identify variables important to wine aroma from a practical, practitioner point of view. This underpins the novelty of the present research on Xinomavro PDO wine, with chemical and sensory correlations needed to improve understanding of the drivers of wine style and typicality based on its chemical components. The current study, therefore, aimed to describe the aroma profile of Xinomavro PDO wines, produced from different vineyards and vinified with a consistent process, by correlating the results from descriptive sensory analysis with quantitative GC-MS data and assessing the impact of different volatile compounds according to their odor activity value (OAV). Additionally, GC-O was applied to wine extracts to identify and rank volatile compounds associated with specific aroma descriptors that may typify the aroma of Xinomavro PDO wine.

## 2. Results and Discussion

### 2.1. Classical Analyses

Table 1 summarizes the basic chemical composition of the Xinomavro research wines produced in this study. Wines had fermented to dryness and ranged in alcohol strength from 12.9 to 13.9% (*v*/*v*), highlighting that the grapes were harvested at technological maturity [3]. Titratable acidity (TA) ranged between 6.6 and 7.9 g/L (expressed as tartaric acid), in agreement with Koussissi et al. [5], and pH values were between 3.31 and 3.51. Volatile acidity (VA) ranged from 0.38 to 0.49 g/L, with values falling well within the legal limit. Regarding color intensity, wines presented low values between 2.3 and 5.9, reinforcing the fact that Xinomavro wines are poor in anthocyanins [1,6]. All parameters presented statistically significant differences between the wine samples. As the winemaking protocol was standardized, this indicates that climatic, pedological, and topographical variations between the test sites could be the cause [14].

### 2.2. Volatile Composition of Xinomavro Wines

A total of 25 volatile compounds were quantified by GC-MS in the Xinomavro research wines (Table 2). These compounds were grouped into five chemical classes, including esters (10), higher alcohols (6), terpenes (4), volatile fatty acids (4), and phenols (1). A comparison of the sum for the total concentrations of each chemical group is shown in Figure 1.

Esters were the most numerous class and comprised 40% of the number of identified compounds. Generally, they contribute to the fruity aroma [17], mainly in the red berry and blackberry spectrum [10]. More specifically, the ester class is represented by two different groups, i.e., ethyl and acetate esters (Figure 1). The total concentration of ethyl esters ranged between 2.2 and 3.1 mg/L (Figure 1A), while the acetate esters total ranged between 0.8 and 1.1 mg/L (Figure 1B). Wine samples in both ester groups presented statistically significant differences: for ethyl esters, Xin4 and then Xin5 were highest, and Xin1 was lowest; for acetate esters, Xin5 was higher than Xin2 but not Xin3 and Xin4, and Xin1 and Xin6 were lowest. Some of these esters are known to be important volatile compounds in red wine [10]. In terms of concentration, the most abundant esters in the Xinomavro research wines were ethyl octanoate (1200–1880 μg/L) and ethyl decanoate (521–764 μg/L), followed by 3-methylbutyl acetate (325–640 μg/L) and hexyl acetate (206–317 μg/L; Table 2).

Higher alcohols are considered to strongly affect overall wine aroma [18]. They are typically found at levels well above their odor threshold and impart “vinous” or “fermented” characters [19]. They could influence the overall aroma by masking other wine odors or contributing to aroma complexity [10]. Alcohols comprised the second most numerous class of volatiles at 24% of the total, but was the major class by far regarding total concentration, ranging from 392 to 466 mg/L (Figure 1C). Significant differences were seen among the six wines, with Xin1 and Xin3 recording the highest values and Xin6 the lowest. 3-Methyl-1-butanol, 2-methyl-1-propanol, and 2-phenylethanol were the major compounds in these wines (Table 2). Xin1 was the highest for 2-phenylethanol, not statistically different from Xin2 and Xin3 for 3-methyl-1-butanol, and the lowest for 2-methyl-1-propanol. In contrast, Xin3 was highest in 2-methyl-1-propanol, and Xin5 and Xin6 were the lowest for 3-methyl-1-butanol and 2-phenylethanol.

Terpenes are primary volatile compounds originating from the grape [20]. They can be released into the wine in their free and/or glycosylated form and they generally improve the varietal aroma character of wines [21]. Four terpenes were identified and quantified—the monoterpenoids linalool, nerol, and geraniol, and the C_13_-norisoprenoid β-damascenone (Table 2)—with the total concentration varying between 0.17 and 0.25 mg/L (Figure 1D). Despite being present at low concentrations, they have considerable impact on the odor properties of wines with their floral and fruity characters [21]. Wine Xin3 was higher in total concentration of terpenes than other wines, except for Xin5, which was no different from the others (Figure 1D). Differences among these compounds can be associated with certain cultivars and regions [21]. β-Damascenone was the only C_13_-norisoprenoid compound quantified, but is worth noting because of its ability to enhance fruity aromas attributed to ethyl esters and mask herbaceous methoxypyrazine-related characters [21]. The Xinomavro wines varied in β-damascenone concentration between 3.4 to 7.3 μg/L (Table 2), in accord with other young red wines [21]. There were statistically significant differences among the wines regarding the concentration of β-damascenone, with Xin3 being the highest and Xin4 the lowest.

Four volatile fatty acids were quantified, with variation in the total concentration from 5.2 to 8.5 mg/L (Figure 1E). There was a statistically significant difference between wines, with sample Xin3 being the highest and Xin4 the lowest. Organic acids have been described with unpleasant aroma notes, but they may only have a small impact on wine aroma [22] and can serve as reactants in the production of desirable ethyl esters [23]. Hexanoic acid and 2-methylpropanoic acid were generally more abundant in the examined wines (Table 2). Xin2 recorded the highest value for 2-methylpropanoic acid and Xin4 the lowest. At the same time, hexanoic acid concentration was lower in Xin4 than Xin3, but not different from Xin5 and Xin6.

Volatile phenols are another class of grape and wine volatiles, with eugenol being solely identified in the Xinomavro wines and representing total concentrations between 81.1 to 138 μg/L (Table 2). Eugenol has been found in grape stems and could be extracted into wine [24]. Notably, Xinomavro is a grape variety with very soft bunches and during destemming and crushing grapes, parts of the stem can end up in the winemaking pomace after being broken into pieces. This could explain the presence of eugenol in these research wines. With an odor described as clove, eugenol has been positively correlated with wine aroma quality [24]. Eugenol concentration differed significantly among the Xinomavro wines, with Xin2, Xin3, and Xin6 having higher values than Xin1, Xin4, and Xin5 (Figure 1F).

### 2.3. Sensory Descriptive Analysis of the Wine

A trained sensory panel described the aroma characteristics of the six Xinomavro wines produced in this study (Figure 2 and Figure 3). The ANOVA results (Appendix A) indicated significant differences among the aroma intensity of the wines and mean scores for cherry, vanilla, olive, sour cherry, tomato, strawberry, berry fruit, mushroom, green bell pepper, plum, and spices (Figure 2). Wine sample Xin4 had the highest total aroma intensity. Wine sample Xin2 presented the highest mean score for the mushroom descriptor, but did not significantly differ from Xin3 for plum (Appendix A). The score for these attributes was quite low (<2), which indicated that they are not sensory attributes that could characterize the varietal aroma profile. Xin3 presented the highest value in the cherry attribute, while Xin1 was the highest in strawberry and berry fruit attributes. For the tomato attribute, the highest value was presented in Xin6, followed by Xin5 and Xin4, with no significant differences between them, whereas green bell pepper was most intense in sample Xin6. Wine samples Xin3, Xin4, and Xin2 scored higher in the spices attribute than Xin1 and Xin5, but did not differ from Xin6. All the Xinomavro wines presented almost zero vanilla aroma intensity, although there was a significant difference among the samples (Figure 2 and Appendix A).

A PCA biplot of the sensory attributes helped visualize the results described above, with the research wines separating into three groups (Figure 3). The first group includes Xin1 and Xin2, which were generally characterized by fruity aromas, including strawberry, cherry, sour cherry, berry fruit, and plum. The other group consists of Xin3, Xin4, and Xin5, which were related to the vegetative group of sensory attributes (tomato, green bell pepper, and olive), along with vanilla. On the contrary, wine samples Xin3 and Xin4 were less influenced by berry fruit and strawberry attributes. Wine sample Xin6 was associated with spices, mushroom, and cherry descriptors and overall aroma intensity, but presented low correlation with berry fruit and strawberry descriptors. These results agree with the sensory profile of Xinomavro wines as presented by the official website of “Wines of Greece” [25]. According to the description, Xinomavro young red wines are characterized by red fruit, spices, and vegetal aromas.

### 2.4. Odor Activity Values

Based on quantitative GC-MS data and published odor detection thresholds, the calculation of odor activity values (OAVs) provides guidance about the most potent odorants for a given product. OAVs are frequently used to indicate the relative importance of volatile compounds and provide insight into the compounds having the greatest influence on aroma [26]. Generally, odorants presenting high OAVs will have a stronger impact on the overall aroma, although a compound with OAV > 1 can still be insignificant in wine because of masking effects [22].

Table 3 shows that 16 out of 25 quantified aroma compounds can be found in Xinomavro wines with OAV > 1. Six compounds, namely 3-methyl-1-butanol, 3-methylbutyl acetate, ethyl octanoate, 3-methylbutyric acid, β-damascenone, and eugenol, were the main compounds (OAV > 10) and could be the principal contributors to the sensory profile of the wines. 3-Methyl-1-butanol likely has an indirect impact on the aroma profile of the wine by affecting the perception of other aroma compounds [18]. 3-Methylbutyl acetate and ethyl octanoate are volatile compounds that were previously detected in red wine from Greek varieties [7] and are associated with fruity aromas [14,18].

Four ethyl esters also had OAV > 1 (Table 3): ethyl hexanoate (≈2–3), ethyl decanoate (≈3), ethyl butanoate (≈2–6), and ethyl 2-methylbutanoate (≈2–3). Wine sample Xin4 had the highest OAVs for ethyl butanoate, ethyl hexanoate, and ethyl decanoate, and the highest score for aroma intensity according to sensory evaluation (Figure 2). The family of esters influence pleasurable fruity aromas in young red wines, while the branched-chain ethyl 2-methylbutanoate imparts red fruit aromas [18]. Wine samples Xin1 and Xin3 presented the highest OAVs in ethyl 2-methylbutanoate. At the same time, they had the highest intensity of strawberry and cherry attribute, respectively (Figure 2). 2-Phenylethanol, 2-methyl-1-propanol, and 3-(methylthio)-1-propanol were also in the group of aroma compounds with OAV > 1. The OAVs for β-damascenone and linalool were ≈7–15 and ≈2–5, respectively. Terpene aroma compounds have also been detected in other red PDO Greek wines [7]. Wine sample Xin3 had the highest OAV for β-damascenone (Table 3), an aroma compound that appears to act as an enhancer of fruity notes in red wines [21], and the highest score in cherry attribute according to sensory evaluation results (Figure 2).

In the case of less desirable aromas such as those from volatile fatty acids, 3-methylbutyric, butanoic, and hexanoic acids had OAVs in the range 4–67. 3-Methylbutyric acid was an order of magnitude higher than the other volatile fatty acids, with an OAV ≈ 31–67 among the wines, followed by butanoic acid with an OAV ≈ 5–8, while hexanoic acid presented a lower OAV ≈ 4–6 (Table 3). Volatile fatty acids are fermentation-derived compounds [10], which at a total concentration over 50 mg/L, could negatively affect the fruity character of the wine. Their total concentration of 5.2–8.5 mg/L (Table 2) was such that they were likely to be positive contributors to quality in the present case, increasing the complexity of the wines [32].

The Xinomavro research wines presented a high content of eugenol (OAV ≈ 14–23), which from a sensory point of view could have an impact by providing spicy aromas [24]. In contrast, a range of the volatile compounds measured did not seemingly contribute to the aroma of these Xinomavro wines, based on their low OAV (Table 3). However, compounds with OAV < 1 could still potentially enhance some of the impact aroma characters through synergistic interactions with other matrix volatiles [18].

It is worth mentioning that the aroma compounds with the greatest influence (OAVs > 1) presented differences among the wine samples produced from different vineyards within the PDO zone. Wine sample Xin3 tended to have the highest OAVs for most aroma compound families, such as the terpenes, higher alcohols along with Xin1, ethyl esters with Xin4, and volatile fatty acids with Xin2, whereas Xin5 presented the highest OAVs in the acetate ester group. This phenomenon indicates that despite the common vinification protocol followed, differences could still exist among the wines due to location (terroir) and meso-climate of subregions within the PDO zone [26], as alluded to earlier. Further investigation is required, however, to examine for such variation within the PDO zone.

### 2.5. Linking Chemical and Sensory Data of Xinomavro Wines

PLSR multivariate analysis was applied to visualize the association between the orthonasal sensory attributes and the OAVs of the volatile aroma compounds (OAV > 1). This approach has been previously used in the modeling of the aroma of wines [33,34,35,36] and distillates [37]. To build the PLS model, two components were chosen, and based on VIP scores > 0.8, 16 of the 17 OAVs > 1 calculated in the present work were analyzed. The two PLS components explained 65% of the total variance in the x-matrix and 65% for the y-matrix. Figure 4 highlights the differences and similarities between the studied wine samples in relation to the composition of volatiles and sensory traits that were determined. Xinomavro wine samples were separated into four groups according to the PLSR components, with Xin1, Xin2, and Xin3 being found to the left along component 1 (generally fruity spectrum) and the remaining wines to the right (greater intensity, green, and vegetal). Component 2 resolved cherry and spice for Xin3 at the top from berry traits associated with Xin1 and Xin2 below, as well as Xin6 and green bell pepper character in the lower quadrant from Xin4 and Xin5 above.

More specifically, the upper left side of Figure 4 contained Xin3, which was characterized by greater intensity of cherry, mushroom, and spices attributes and volatiles such as ethyl 2-methylbutanoate, 3-(methylthio)-1-propanol (methionol), isoprenoids (linalool and β-damascenone), and several fatty acids. Xin1 and Xin2 appeared in the lower left side and were related to strawberry, sour cherry, and berry fruit attributes, along with two higher alcohols (2-phenylethanol, 3-methyl-1-butanol). Xin4 and Xin5 were located in the upper right side of Figure 4 and were mainly associated (especially Xin4) with aroma intensity, as well as olive, vanilla, and tomato traits, in conjunction with a range of esters, including ethyl hexanoate, ethyl octanoate, ethyl butanoate, and 3-methylbutyl acetate, which related to the aroma intensity of the studied wines. Xin6 was mostly associated with the green bell pepper attribute, located on the lower right side of the plot, as well as with a lack of the terpene group aroma compounds (Figure 4).

### 2.6. Olfactometric Data of Xinomavro Wines

GC-O was conducted to characterize the odor active zones of Xinomavro research wines. The selection of samples for solvent-assisted flavor evaporation (SAFE) and GC-O analysis was based on their different aroma characteristics, as revealed in the PCA biplot (Figure 3). The findings of the olfactometric analysis are summarized in Table 4. The identity of the majority of the odorants was confirmed by matching retention index, mass spectrum, and odor quality to authentic standards. The aroma active volatiles defining these Xinomavro wines were esters (12), higher alcohols (4), C6 compounds (2), volatile fatty acids (6), sulfur-containing compounds (3), aldehydes (3), a methoxypyrazine (1), furanones (3), lactones (2), terpenes (3), and a phenol (1). As commonly occurs, there was a positive correlation between the MF value of the aroma compounds and the perception of their odors [12]. The main compounds and associated odors with MF value >70% were: ethyl butanoate, imparting red fruit odor; 3-methyl-1-butanol, solvent like; ethyl hexanoate, fruity; 1-hexanol and (Z)-3-hexenol, grass; 3-isobutyl-2-methoxypyrazine, green bell pepper; 2-methylpropanoic acid, cheesy; 3-methylbutyric and hexanoic acids, sweat-like; dimethylmethoxyfuranone, strawberry; furaneol and homofuraneol, cotton candy/caramel; ethyl decanoate, grape; methionol, mushroom; β-damascenone, fresh/canned fruit; 2-phenylethanol, rose; and sotolon, curry (Table 4). The majority of the aroma-active compounds detected in the wines were those that provided strong fresh fruit (especially red fruit), grass, spices, and canned/dried fruit odors. As also determined by the sensory evaluation (Figure 2 and Figure 3), most of these aromas were used to characterize the sensory attributes of the Xinomavro wines. Results from the quantitative analysis also supported these outcomes.

From another point of view, it is important to recognize that GC-O analysis differs from classical sensory analysis because of the chromatographic separation of components, whereas mixtures of odorants present in the original matrix can have a synergistic effect in sensory analysis [10]. Notably, the ability of the human brain to perceive discrete odors is limited when dealing with complex mixtures [38]. In such mixtures, the brain normally recognizes a general attribute common to the groups of odorants, such as “fruity” or “spicy”. As such, the odorants in Table 4 were grouped according to their generic aroma properties as follows: F, odorants with fruity character; V, vegetal character; S, spicy/dried/canned fruit notes; ST, strawberry/sweet character; and B, berry fruit character. Pearson correlation analysis revealed that the summed scores of all odorants possessing a strawberry/sweet character according to GC-O was correlated strongly (r = 0.85, *p* = 0.036) with the score for strawberry attribute arising from sensory evaluation of the wines. Likewise, the summed scores of compounds with spicy character from GC-O was correlated well (r = 0.91, *p* = 0.044) with the wine sensory scores for the spicy character. The sum of fruity odorants from GC-O correlated well (r = 0.75, *p* = 0.041) with the berry fruit intensity of the wines, with the summation of the three esters that presented OAV > 10 according to the quantitative data being correlated even more strongly (r = 0.98, *p* = 0.023) with the berry fruit sensory score. There may also be a correlation between the summed odorants with vegetal character and the green bell pepper sensory attribute of the wines, although the relationship was barely significant (r = 0.59, *p* = 0.046).

The strawberry/sweet group included three aroma odorants, i.e., dimethylmethoxyfuranone, furaneol, and homofuraneol. It has been reported that relatively small quantities of furaneol, by acting in a synergistic manner with homofuraneol, can exert a great impact on the strawberry and caramel notes of wines [39,40]. Dimethylmethoxyfuranone is an enzymatic methylation product of furaneol that has been correlated with strawberry aroma [41]. Concluding, these aroma compounds belong to the primary aroma arising directly from the grapes [8]. Therefore, it could be proposed that the typical strawberry aroma of young Xinomavro PDO wines is the result of the synergistic effect of these three aroma compounds and belongs to the primary odorant class.

The fruity group included ethyl and acetate ester odorants. Esters contribute to the fruity character of young wines [17], even at subthreshold levels [10]. This phenomenon could explain why only the summation of ester odorants correlated well with the berry fruit attribute and not the individual esters. Decreasing the number of esters of the fruity group according to the quantitative results created another group named berries, which had a very strong correlation between these odorants and the berry sensory character. Esters are mainly enzymatically synthesized by yeast during alcoholic fermentation and their contents can also be modulated by lactic acid bacteria during malolactic fermentation. It is well known that esters are generally produced in excess by yeast, which are mainly responsible for the corresponding ester composition of the wines [42], so they contribute to the secondary aroma produced by fermentation [8]. Although the contributions of grapes to the formation of ethyl and acetate esters from fermentation is not fully elucidated, Ferreira et al. [27] have indicated that esters could affect the varietal aroma of young red wines. Ester concentration ultimately appears to be a combination of the precursors in a grape juice or must and winemaking variables [43,44,45].

The last group of odorants that correlated well with the sensory attribute was the spicy group consisting of β-damascenone, eugenol, and sotolon. Sotolon is a lactone that can be found in some young red wines, but is more clearly found in aged wines, which could indicate that it forms during wine maturation [22]. β-Damascenone is usually associated with fruity and floral aromas, but it has also been associated with caramel [46] and canned fruits [39]. As described by the GC-O panelists, its aroma in the present study more closely resembled the spicy/dried fruit group. This C_13_-norisoprenoid varietal aroma compound, found in grapes and the respective wines, is derived from the breakdown of different aglycones and neoxanthin from grape [21]. The last odorant is eugenol, a volatile phenol that could be extracted into the wine [24], as mentioned earlier.

## 3. Materials and Methods

### 3.1. Wine Samples

#### 3.1.1. Winemaking Protocol

Xinomavro grapes were hand harvested at technological maturity (Appendix A) during the 2020 vintage from six vineyard plots located in the Amyndeon PDO region (Appendix A). The plots were selected to provide broad coverage of the region’s major soil and microclimate conditions and potentially reflect distinctive sensory and chemical differences in Xinomavro wines. Harvested grape parcels underwent small-scale vinification at the Laboratory of Enology and Alcoholic Drinks using a common protocol. A total of 75 kg of Xinomavro grapes from each vineyard was separated into three portions of 25 kg, and each was destemmed and crushed to provide musts for triplicate fermentation. The grape musts were transferred to 30 L stainless steel tanks for vinification, 40 mg/L of sulfur dioxide was added, and after 4 h, tanks were inoculated at a rate of 250 mg/L of *Saccharomyces cerevisiae* strain HDS 135 (Fermentis, France). Then, 12 h after yeast addition and again after fermentations was 1/3 completed, 125 mg/L of diammonium phosphate (DAP) and 125 mg/L of Springferm (Fermentis, France) were added. Maceration was conducted by pumping the juice over twice a day and alcoholic fermentation was performed between 21–23 °C, until sugar depletion, as determined using an enzymatic analyzer (Y15, Biosystems S.A., Barcelona, Spain). Once dry, ferments were pressed with the use of a hydraulic press (Pew80, Grifo Marchetti, Piadena Drizzona, Italy) to separate the wine from skin and seed. The wines were transferred to 30 L stainless steel tanks and inoculated with Viniflora CH11 (CHR Hansen, Hørsholm, Denmark) for malolactic fermentation (MLF). Once the malic acid concentration was below 0.2 g/L, as determined enzymatically, MLF was considered complete. After racking and addition of 50 mg/L of sulfur dioxide, wines were stabilized in a cold room (~5 °C) for a period of 2 months. Amber glass bottles of 0.75 L were filled, the headspace was flushed with N_2_, and the bottles were closed with a technical stopper (45 mm length, Diam 5). Bottled wines were stored horizontally in cellar conditions (15 ± 2 °C, 55–60% humidity). Wines produced in this manner were labeled as Xin1, Xin2, Xin3, Xin4, Xin5, and Xin6 according to the respective vineyard, with the three fermentation replicates for each wine bottled separately.

#### 3.1.2. Basic Wine Composition

Analysis of enological parameters and determination of chromatic characteristics was undertaken after vinification. Reducing sugars, alcoholic degree, TA, pH, and VA were determined according to OIV methods [47]. Color intensity was determined as the sum of absorbances at 420, 520, and 620 nm in a 1 cm pathlength cell [48], using a spectrophotometer (UV-1900, Shimadzu Scientific Instruments, Kyoto, Japan). Hue was quantified as the ratio of absorbance at 420 and 520 nm [48]. All measurements were undertaken in triplicate for each winemaking replicate.

### 3.2. Quantification of Wine Volatile Compounds

#### 3.2.1. Isolation of Volatiles for Liquid Injection

The method described by Ivanova et al. [49] for the analysis of wine volatiles was used, with slight modifications. The wine samples (40 mL) were spiked with ethanol solutions of 3-octanol, ethyl heptanoate, and heptanoic acid as internal standards (final concentrations of 10 mg/L), 5 mL of dichloromethane were added, and the mixture was stirred continuously with a magnetic stirrer for 15 min. The organic layer was removed and a second extraction was carried out with an additional 5 mL of dichloromethane. The organic extracts were combined, centrifuged at 1968 rcf for 10 min at 4 °C, and the lower organic layer was transferred into a vial and dried with 1.5 g of anhydrous sodium sulfate. After filtration, the organic extract was concentrated to 500 μL under a nitrogen stream. The extraction was performed in duplicate for each winemaking replicate.

#### 3.2.2. Gas Chromatography–Mass Spectrometry (GC-MS)

Analysis of wine extracts was performed with liquid injection using a Perkin Elmer Clarus 590 gas chromatograph with an HT3000A automatic liquid sampler (HTA S.R.L., Brescia, Italy) coupled to a Perkin Elmer Clarus SQ8S mass spectrometer (Perkin Elmer, Waltham, MA, USA). A DB-WAX column (50 m × 0.20 mm i.d., 0.20 μm film thickness, Agilent J&W, Folsom, CA, USA) was used with helium as carrier gas at a constant flow rate of 1.9 mL/min. Injection volume was 1 µL in splitless mode, the injector temperature was set at 250 °C, and the split vent was opened after 120 s. The oven temperature was held at 40 °C for 2 min, heated at 5 °C/min to 240 °C, and held at 240 °C for 20 min. The transfer line and quadrupole temperatures were both 250 °C. Electron ionization (EI) mass spectra were recorded at 70 eV in the range *m*/*z* 40–400 using selected ion full ion (SIFI) mode. A total of 25 wine volatile compounds were identified and quantified using commercial standards with external calibration curves.

### 3.3. Sensory Evaluation

The sensory assessment was carried out by 12 trained panelists (7 females and 5 males; 25–57 years of age) who gave informed consent to take part in the study. Panelists attended three training and two evaluation sessions over a period of 2 weeks. The training period included an initial session in which the panelists were served a set of representative Xinomavro samples to familiarize themselves with this kind of wine variety. The panelists were asked to describe the perceived aroma attributes, as described by Nanou et al. [50]. The second session involved training of panelists using suitable reference standards, one for each of the 11 olfactory attributes on the list (Table 5).

Training was repeated and the panel’s performance was evaluated, with two final sessions being used to assess the samples (six wines in duplicate), as described by Nanou et al. [50]. Each wine was provided monadically in accordance with a Latin square design, with 1 min break enforced between each sample. The tests were carried out in individual booths with natural lighting. Panelists were provided 30 mL of wine in transparent ISO wine glasses covered with plastic Petri dishes and coded with three-digit random numbers. Winemaking replicates for a given wine were blended in equal proportions and left to reach room temperature (20–22 °C). Compusense20 Cloud software (Compusense, Guelph, ON, Canada) was used for data collection. Sensory attribute intensity was rated according to a seven-point scale (ranging from 1: not perceived to 7: very strong).

### 3.4. Isolation of Volatiles for GC-O

#### 3.4.1. Solvent-Assisted Flavor Evaporation (SAFE) Extract

A 500 mL sample of wine was extracted sequentially using 80, 80, and 50 mL of dichloromethane, with magnetic stirring for 10 min. The collected organic phases were combined and dried over anhydrous sodium sulfate [41]. After filtration, the volatiles were isolated using SAFE glassware (Glasbläserei Bahr, Manching, Germany). Throughout the process, a high vacuum (10^−3^ Pa) was maintained and the apparatus was thermostated at 40 °C. The dichloromethane extract of a sample was introduced dropwise at a rate of 10 mL/min. Vacuum to the apparatus was released 5 min after the last drops of extract were added and the collected distillate was allowed to reach room temperature [51]. This SAFE extract was concentrated to 1 mL by distillation at 40 °C in a water bath using a Kuderna–Danish apparatus with a graduated concentrator tube (Witeg Labortechnik GmbH, Wertheim, Germany). The extract was stored at −20 °C until GC-O analysis.

#### 3.4.2. Gas Chromatography–Olfactometry

Analysis of SAFE extracts was conducted on a Perkin Elmer Clarus 590 gas chromatography coupled to a Perkin Elmer Clarus SQ8S mass spectrometer (Perkin Elmer, Waltham, MA, USA). A Perkin Elmer olfactory detector port (SNFR^TM^ Olfactory Port, N6590100) was fitted to the GC-MS system, connected by a flow splitter to the column exit (Swafer^TM^ technology, Perkin Elmer, Waltham, MA, USA), to achieve a 1:5 split ratio. GC effluent was combined with humidified air (99.999% purity, Revival, Greece) at the bottom of the detector port. Injection of 1 μL sample of each concentrated dichloromethane extract was undertaken in splitless mode (injector temperature 250 °C) and the split vent was opened after 5 min. A DB-Wax column (50 m × 0.25 mm i.d., 0.25 μm film thickness Agilent J&W, Folsom, CA, USA) was used with helium as carrier gas in constant flow rate of 1.9 mL/min. The temperature program was as follows: 40 °C for 2 min, then increasing by 4 °C/min to 240 °C, with a 20-min isothermal phase at 240 °C [41]. The MS transfer line was set at 250 °C and electron ionization mass spectra at 70 eV were recorded in the range of *m*/*z* 40–400.

Four panelists (two females and two males, 27–55 years old) trained in descriptive sensory analysis and able to identify the odors of different wine samples of Xinomavro grape variety were used as sniffers. The panelists noted each odor-active compound’s perceived characteristics, intensity, and retention time as they sniffed via the olfactory mask. The odor intensities were assessed using a three-point discrete intensity scale (1–3) and the modified frequency, MF (%), was calculated [10]:MF (%) = √(F (%) × I (%))
where F (%) is the detection frequency of an attribute expressed as a percentage and I (%) is the average intensity expressed as a percentage of maximum intensity.

The MS data were analyzed with TurboMass Ver6.1.2 GC/MS Software (Perkin Elmer, Waltham, MA, USA) and compared with the NIST 2008 Mass Spectra library (US National Institute of Standards and Technology, Gaithersburg, MD, USA). The Kovats retention index (RI) of the target compound was calculated (using C7–C30 alkanes, Sigma-Aldrich) and compared with a NIST database. Each analyte’s aroma description was compared to the literature and to the Flavornet website [31].

### 3.5. Statistical Analysis

Values are presented as the mean ± standard deviation with statistical analyses being undertaken with JMP Pro Statistical Software (version 13). One-way ANOVA (*p* < 0.05) was conducted for volatile compounds and sensory attributes in the six Xinomavro wines followed by Tukey’s HSD post-hoc test (α = 0.05). Sensory data means were subjected to principal component analysis (PCA). Pearson’s correlation coefficient was calculated to reveal the relationship between the summed scores of the odorants according to GC-O analysis and the sensory attributes. Partial least squares regression (PLSR) analysis (PLS2) was conducted using XLSTAT (version 2018, Addinsoft, New York, NY, USA) with the volatile compounds having OAV > 1 (x-variables) and sensory DA data (y-variables). The optimal number of components for the PLSR model was determined from the variable importance in projection (VIPs) plots, including only the variables with VIP > 0.8.

## 4. Conclusions

This study investigated the chemical composition, aroma-active compounds, and sensory characters of Xinomavro red wines arising from six vineyards within the Amyndeon PDO zone. Correlations between the chemical and sensory results were undertaken and the experiments highlighted a clear relationship between the summed groups of aroma-active odorants that were identified via GC-O and the scores from sensory descriptive analysis. The strongest correlations were presented between the strawberry/caramel group and the strawberry attribute, along with the spicy group with the spices attribute, followed by the fruity group with the berry fruit attribute. According to the quantitative results, the quantified esters of the fruity group presented OAVs > 1, whereas the esters of the berries group, and β-damascenone and eugenol of the spicy group, presented OAVs >10. PLSR analysis effectively distinguished the three different groups of Xinomavro wines based on the aroma characters—fruity, spice, and vegetative/green—which were associated with certain aroma compounds. As the first study of its kind, this work has provided strong indications regarding the aroma compounds defining the sensory characteristics of strawberry, berries and spices that may typify Xinomavro PDO wines. Most of these aroma compounds contribute to the primary aroma of the wines, whether arising from the terpenic family and released in the wine after glycoside hydrolysis or some others, such as the acetate and ethyl esters, produced during the vinification process derived from the precursors of the grape must.

Armed with this knowledge, winemakers could use suitable vinification protocols, altering winemaking variables such as yeast strain and/or fermentation temperature, to produce a wine with a prototypical wine style. Manipulation of the wine style could also potentially start in the vineyard through understanding the viticultural practices affecting the formation of precursors in the grape berry. As such, viticultural practices could be adjusted to produce grapes that contain either higher or lower concentrations of precursors to fit the desired wine style that is characteristic of the PDO wine.

While the primary focus of this initial research was the correlations between quantitative GC-MS, GC-O, and sensory data, the work provides opportunities for additional methodologies that can improve and supplement the results of this study. For instance, research could also focus on the identification of the aroma compounds related to the tomato character of Xinomavro PDO wines, considering that the perceived intensity of the tomato attribute was quite high, but no correlation was presented with the vegetal group of odorants.

## Figures and Tables

**Figure 1 molecules-28-05016-f001:**
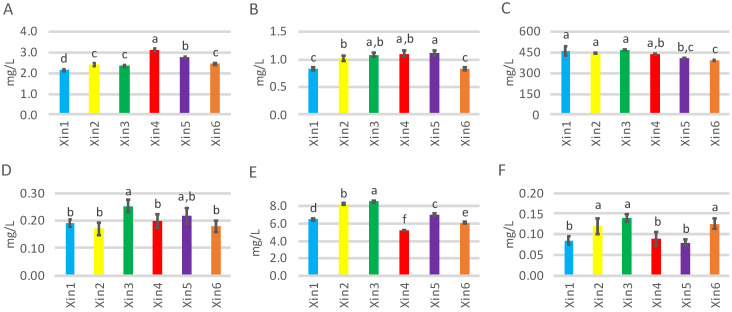
Graphs of the total concentration (mg/L) ± standard deviation of (**A**) ethyl esters, (**B**) acetate esters, (**C**) higher alcohols, (**D**) terpenes, (**E**) volatile fatty acids, and (**F**) volatile phenols for six Xinomavro wines (Xin1–Xin6). Bars with different lowercase letters in the same graph are significantly different according to the post-hoc test (α = 0.05).

**Figure 2 molecules-28-05016-f002:**
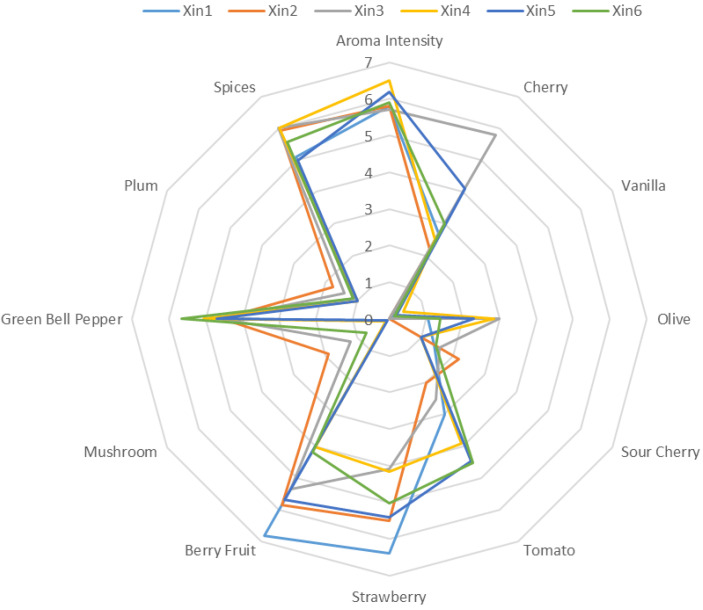
Mean scores of the sensory attributes of the six Xinomavro wines (Xin1–Xin6). There was a significant difference (*p* < 0.0001) among the wines for each of the attributes (Appendix A).

**Figure 3 molecules-28-05016-f003:**
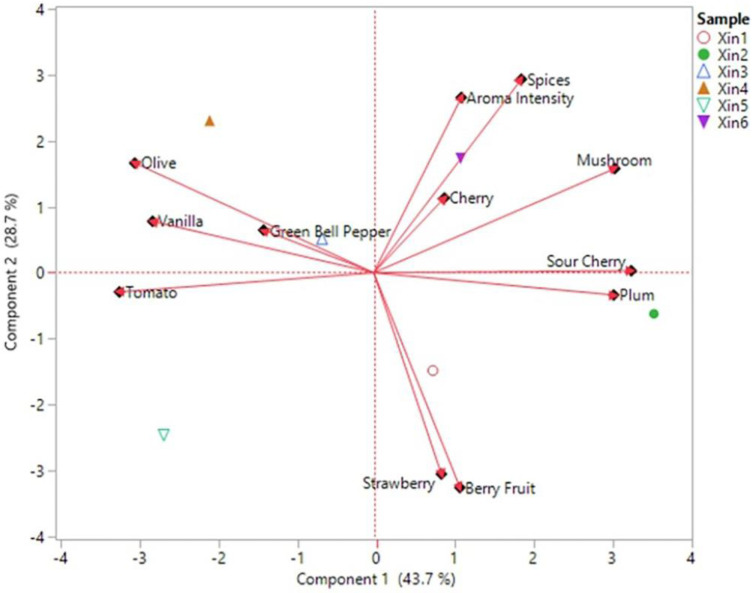
Principal component analysis (PCA) biplot of the sensory attributes of six Xinomavro wines (Xin1–Xin6).

**Figure 4 molecules-28-05016-f004:**
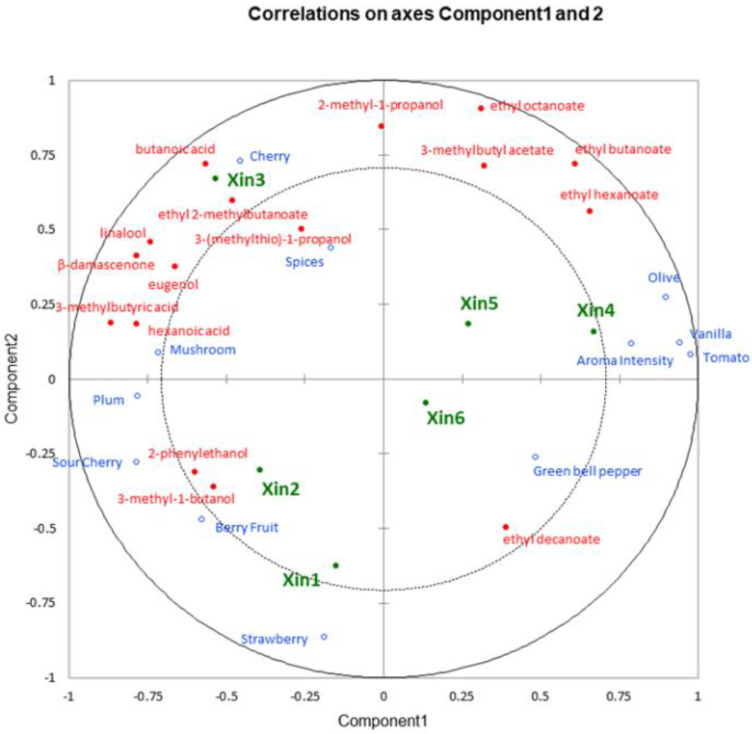
PLSR correlation loadings plot of the OAVs of the volatile aroma compounds as x-variables and scores of the sensory attributes of wine samples as y-variables among the six Xinomavro wines (Xin1–Xin6).

**Table 1 molecules-28-05016-t001:** Mean values (± standard deviation, *n* = 3) of enological parameters and chromatic characteristics of Xinomavro red wines from six vineyards in the Amyndeon PDO zone. Values with different lowercase letters in the same row are significantly different according to the post-hoc test (α = 0.05).

	Xin1	Xin2	Xin3	Xin4	Xin5	Xin6	*p*-Value
Enological Parameters							
Glucose and fructose (g/L)	0.03 ± 0.01 ab	0.02 ± 0.01 c	0.02 ± 0.00 bc	0.02 ± 0.00 bc	0.02 ± 0.00 bc	0.04 ± 0.00 a	0.0004
pH	3.41 ± 0.04 b	3.33 ± 0.01 c	3.31 ± 0.01 c	3.39 ± 0.01 b	3.51 ± 0.00 a	3.33 ± 0.00 c	<0.0001
Titratable acidity (tartaric acid g/L)	6.8 ± 0.1 c	7.7 ± 0.0 b	7.8 ± 0.0 ab	7.9 ± 0.1 a	6.6 ± 0.1 d	7.8 ± 0.0 ab	<0.0001
Volatile acidity (acetic acid g/L)	0.38 ± 0.03 b	0.40 ± 0.00 b	0.42 ± 0.03 ab	0.38 ± 0.02 b	0.43 ± 0.05 ab	0.49 ± 0.04 a	0.0002
Alcohol (% *v*/*v*)	12.9 ± 0.1 d	12.9 ± 0.1 d	13.8 ± 0.1 ab	13.9 ± 0.1 a	13.1 ± 0.1 c	13.7 ± 0.1 b	<0.0001
Chromatic Characteristics							
Color Intensity	2.7 ± 0.1 cd	3.3 ± 0.3 c	4.5 ± 0.5 b	5.9 ± 0.7 a	2.3 ± 0.0 d	3.7 ± 0.0 bc	<0.0001
Hue	0.6 ± 0.0 b	0.6 ± 0.1 b	0.6 ± 0.0 b	0.5 ± 0.1 b	0.7 ± 0.0 a	0.6 ± 0.0 b	0.0013

**Table 2 molecules-28-05016-t002:** Mean concentration (± standard deviation, n = 3) of volatiles determined in Xinomavro red wines from different vineyards (Xin1–Xin6) in the Amyndeon PDO zone. Values with different lowercase letters in the same row are significantly different according to the post-hoc test (α = 0.05).

	Xin1	Xin2	Xin3	Xin4	Xin5	Xin6	*p*-Value
Higher alcohols (mg/L)							
2-Methyl-1-propanol	62.4 ± 1.3 e	83.9 ± 1.2 c	93.7 ± 3.3 a	88.6 ± 0.9 b	85.1 ± 2.6 bc	71.8 ± 0.6 d	<0.0001
3-Methyl-1-butanol	340 ± 33 a	313 ± 3 abc	318 ± 5 ab	301 ± 5 bc	285 ± 2 c	282 ± 6 c	0.0002
1-Hexanol	2.33 ± 0.29 b	2.41 ± 0.16 b	3.58 ± 0.28 a	2.32 ± 0.23 b	2.40 ± 0.27 b	2.30 ± 0.16 b	<0.0001
(*Z*)-3-Hexenol	0.17 ± 0.02 b	0.15 ± 0.01 b	0.21 ± 0.01 a	0.11 ± 0.02 c	0.14 ± 0.02 bc	0.11 ± 0.01 c	<0.0001
3-(Methylthio)-1-propanol	1.34 ± 0.03 cd	2.12 ± 0.08 a	2.22 ± 0.19 a	1.17 ± 0.08 d	1.79 ± 0.07 b	1.50 ± 0.06 c	<0.0001
2-Phenylethanol	53.3 ± 1.5 a	47.4 ± 2.0 b	48.2 ± 0.6 b	41.9 ± 1.2 c	37.3 ± 1.0 d	35.1 ± 1.0 d	<0.0001
Esters (μg/L)							
3-Methylbutyl acetate	325 ± 26 d	436 ± 12 c	529 ± 32 b	549 ± 40 b	640 ± 36 a	357 ± 32 d	<0.0001
2-Methylpropyl acetate	138 ± 17 a	141 ± 18 a	113 ± 6 a	70.9 ± 23 b	63.4 ± 7.8 b	48.1 ± 2.4 b	<0.0001
Hexyl acetate	206 ± 17 c	246 ± 31 bc	256 ± 2.8 b	317 ± 14 a	237 ± 15 bc	247 ± 33 bc	<0.0001
2-Phenylethyl acetate	166 ± 6.4 b	197 ± 19 a	181 ± 4.8 ab	166 ± 17 b	179 ± 6.4 ab	178 ± 12 ab	0.0184
Ethyl octanoate	1200 ± 25 e	1450 ± 49 c	1350 ± 20 b	1880 ± 48 a	1720 ± 28 b	1490 ± 21 c	<0.0001
Ethyl 2-methylbutanoate	51.7 ± 1.8 ab	34.9 ± 6.2 b	61.0 ± 4.2 a	45.5 ± 17 ab	45.1 ± 5.0 ab	46.4 ± 2.3 ab	0.0069
Ethyl 3-methylbutanoate	45.4 ± 3.1 b	61.4 ± 8.2 a	61.5 ± 4.7 a	56.5 ± 2.6 a	52.5 ± 2.9 ab	60.3 ± 4.3 a	0.0009
Ethyl hexanoate	128 ± 11 c	125 ± 9.4 c	275 ± 18 a	245 ± 4.1 b	235 ± 16 b	231 ± 12 b	<0.0001
Ethyl decanoate	687 ± 28 bc	734 ± 16 ab	521 ± 19 d	764 ± 21 a	635 ± 35 c	540 ± 11 d	<0.0001
Ethyl butanoate	37.4 ± 7.9 c	36.0 ± 11 c	89.5 ± 20 b	120 ± 6.3 a	74.5 ± 9.9 b	81.3 ± 15 b	<0.0001
Fatty acids (mg/L)							
Hexanoic Acid	2.15 ± 0.15 b	2.11 ± 0.08 b	2.45 ± 0.13 a	1.81 ± 0.08 c	1.76 ± 0.03 c	1.82 ± 0.14 c	<0.0001
2-Methylpropanoic acid	2.08 ± 0.01 d	2.76 ± 0.04 a	2.61 ± 0.04 b	1.46 ± 0.02 f	2.35 ± 0.04 c	1.54 ± 0.03 e	<0.0001
Butanoic acid	0.94 ± 0.03 d	1.16 ± 0.04 bc	1.42 ± 0.05 a	0.95 ± 0.02 d	1.25 ± 0.03 ab	1.01 ± 0.17 cd	<0.0001
3-Methylbutyric acid	1.34 ± 0.02 d	2.21 ± 0.02 a	2.02 ± 0.08 b	1.02 ± 0.06 e	1.70 ± 0.03 c	1.73 ± 0.02 c	<0.0001
Terpenes (μg/L)							
Linalool	88.5 ± 13 b	62.5 ± 14 c	132.0 ± 17 a	78.4 ± 10 b,c	70.9 ± 8.9 bc	67.8 ± 7.9 bc	<0.0001
Nerol	64.8 ± 9.5 b	69.8 ± 12 b	75.3 ± 20 b	82.7 ± 16 ab	110 ± 16 a	72.4 ± 11 b	0.0054
Geraniol	33.1 ± 3.8 ab	29.4 ± 2.3 b	36.1 ± 2.1 a	32.3 ± 3.3 b	31.5 ± 1.5 b	32.4 ± 1.5 b	0.0009
β-Damascenone	4.50 ± 0.70 bc	5.90 ± 0.80 ab	7.30 ± 1.10 a	3.40 ± 0.35 c	5.70 ± 0.80 ab	6.10 ± 0.90 ab	<0.0001
Volatile phenols (μg/L)							
Eugenol	85.9 ± 10 b	122 ± 18 a	138 ± 8.2 a	89.2 ± 16.3 b	81.1 ± 8.20 b	125 ± 12.9 a	<0.0001

**Table 3 molecules-28-05016-t003:** Odor activity values (OAVs) of odorants found in the six Xinomavro red wines from different vineyards (Xin1–Xin6) in the Amyndeon PDO zone.

	Odor Threshold (mg/L) ^a^	Sensory Descriptor ^b^	OAV ^c^					
			Xin1	Xin2	Xin3	Xin4	Xin5	Xin6
Higher alcohols								
2-Methyl-1-propanol	40 [27]	wine, solvent, bitter	**1.6**	**2.1**	**2.3**	**2.2**	**2.1**	**1.8**
3-Methyl-1-butanol	30 [28]	whiskey, malt, burnt	**11**	**10**	**10**	**10**	**9.5**	**9.3**
1-Hexanol	8 [27]	resin, flower, green	0.3	0.3	0.5	0.3	0.3	0.3
(*Z*)-3-Hexen-1-ol	0.04 [27]	grass	0.4	0.4	0.3	0.3	0.3	0.5
3-(Methylthio)-1-propanol	1 [27]	sweet, potato	**1.3**	**2.1**	**2.2**	**1.2**	**1.8**	**1.5**
2-Phenylethanol	14 [27]	honey, spice, rose, lilac	**3.8**	**3.4**	**3.5**	**3.0**	**2.7**	**2.5**
2-Methyl-1-propanol	40 [27]	wine, solvent, bitter	**1.6**	**2.1**	**2.3**	**2.2**	**2.1**	**1.8**
Esters								
3-Methylbutyl acetate	0.03 [27]	banana	**11**	**15**	**18**	**18**	**21**	**12**
2-Methylpropyl acetate	1.6 [22]	fruit, apple, banana	0.1	0.1	0.1	0.0	0.0	0.0
Hexyl acetate	1.5 [29]	fruit, herb	0.1	0.2	0.2	0.2	0.2	0.2
2-Phenylethyl acetate	0.25 [28]	rose, honey, tobacco	0.7	0.8	0.7	0.7	0.7	0.7
Ethyl butanoate	0.02 [28]	apple	**1.9**	**1.8**	**4.5**	**6.0**	**3.7**	**4.1**
Ethyl 2-methylbutanoate	0.018 [27]	apple	**2.9**	**1.9**	**3.4**	**2.5**	**2.5**	**2.6**
Ethyl 3-methylbutanoate	0.03 [27]	fruit	0.0	0.1	0.1	0.1	0.1	0.1
Ethyl octanoate	0.014 [27]	apple peel, fruit	**9.2**	**8.9**	**20**	**18**	**17**	**17**
Ethyl hexanoate	0.58 [28]	fruit, fat	**2.1**	**2.5**	**2.6**	**3.3**	**3.0**	**2.3**
Ethyl decanoate	0.2 [27]	grape	**3.5**	**3.7**	**2.6**	**3.8**	**3.2**	**2.7**
Fatty acids								
Hexanoic acid	0.42 [27]	sweat	**5.1**	**5.0**	**5.8**	**4.3**	**4.2**	**4.3**
2-Methylpropanoic acid	8.1 [27]	rancid, butter, cheese	0.3	0.3	0.3	0.2	0.3	0.2
Butanoic acid	0.173 [27]	rancid, cheese, sweat	**5.4**	**6.7**	**8.2**	**5.5**	**7.2**	**5.8**
3-Methylbutyric acid	0.033 [27]	sweat, acid, rancid	**40**	**67**	**61**	**31**	**51**	**58**
Terpenes								
Linalool	0.025 [27]	flower, lavender	**3.5**	**2.5**	**5.3**	**3.1**	**2.8**	**2.7**
Nerol	0.5 [30]	sweet	0.1	0.1	0.2	0.2	0.2	0.1
Geraniol	0.036 [20]	rose, geranium	0.9	0.8	1	0.9	0.9	0.9
β-Damascenone	0.00005 [28]	apple, rose, honey	**9.0**	**12**	**15**	**6.9**	**11**	**12**
Phenols								
Eugenol	0.006 [27]	clove, honey	**14**	**20**	**23**	**15**	**14**	**21**

^a^ Reference given in parentheses. In refs. [20,27], the matrix was a 10% water/ethanol solution at pH 3.2. In ref. [28], the mixture was 10% in ethanol. In ref. [29], thresholds were calculated in a 12% water/ethanol mixture ethanol. In ref. [20], thresholds were calculated in a 10% water/ethanol mixture containing 5 g/L of tartaric acid at pH 3.2. In ref. [30], the matrix was a 40% ethanol/water (*v*/*v*) mixture. ^b^ The aroma description of each analyte was obtained from the Flavornet website [31]. ^c^ OAVs shown in bold represent the odor active volatile compounds (OAV > 1).

**Table 4 molecules-28-05016-t004:** Odor description, compound identification, aroma group designation, and modified frequency (MF (%)) of the odor-active compounds in Xinomavro research wines determined by GC-O. Compounds in bold represent the most powerful odor active volatiles identified in one or more wines (MF > 70%).

No.	RI	Odorant Description ^a^	Identity Determined by ^b^	Compound	Group Code ^c^	MF (%)			
						Xin1	Xin4	Xin5	Xin6
1	997	Fruity, sweet	*MS*, RI, O	2-methylpropyl acetate	F	58	- ^d^	-	-
2	1020	Red fruit	*MS*, RI, O	**ethyl butanoate**	F	47	82	82	47
3	1049	Apple	*MS*, RI, O	ethyl 2-methylbutanoate	F	33	47	33	33
4	1064	Fruity	*MS*, RI, O	ethyl 3-methylbutanoate	F	47	33	47	33
5	1117	Banana	*MS*, RI, O	3-methylbutyl acetate	F,B	67	33	33	58
6	1156	Wine	RI, O	1-butanol		-	67	67	-
7	1210	Burnt, solvent	*MS*, RI, O	**3-methyl-1-butanol**		100	100	100	100
8	1240	Fruity	*MS*, RI, O	**ethyl hexanoate**	F,B	47	33	67	82
9	1279	Fruit, herb	*MS*, RI, O	hexyl acetate	F	58	47	67	67
10	1309	Burnt	*MS*, RI, O	2-methyl-3-furanthiol		47	47	33	67
11	1356	Resin, flower, grass	*MS*, RI, O	**1-hexanol**	V	67	82	82	82
12	1379	Fat	MS, RI, O	nonanal		47	67	33	33
13	1391	Grass	*MS*, RI, O	**(*Z*)-3-hexen-1-ol**	V	47	82	82	67
14	1433	Fruit	*MS*, RI, O	ethyl octanoate	F,B	47	33	67	58
15	1442	Sour	MS, RI, O	acetic acid		82	82	82	82
16	1446	Sweet, bread	MS, RI, O	furfural		47	47	47	-
17	1466	Cooked potato	MS, RI, O	3-(methylthio)propanal (methional)		58	58	33	33
18	1516	Green pepper	RI, O	**3-isobutyl-2-methoxypyrazine**	V	33	33	82	67
19	1533	Flower	*MS*, RI, O	linalool		58	58	58	58
20	1553	Butter, cheese	*MS*, RI, O	**2-methylpropanoic acid**		47	82	67	58
21	1584	Cotton candy, strawberry	MS, RI, O	**2,5-dimethyl-4-methoxy-3(2*H*)-furanone (dimethylmethoxy furanone)**	ST	47	82	58	33
22	1607	Rancid, cheese	*MS*, RI, O	butanoic acid		58	58	58	58
23	1640	Grape, flower	*MS*, RI, O	**ethyl decanoate**	F	58	82	58	47
24	1647	Honey	MS, RI, O	γ-butyrolactone		-	47	-	67
25	1651	Sweat, rancid	*MS*, RI, O	**3-methylbutyric acid**		82	100	58	58
26	1718	Potato, mushroom	*MS*, RI, O	**3-(methylthio)-1-propanol (methionol)**		82	58	100	82
27	1781	Fruit, sweet, rose	MS, RI, O	citronellyl butyrate	F	47	58	33	33
28	1816	Fresh-canned fruit	*MS*, RI, O	**β-damascenone**	S	33	100	67	47
29	1833	Sweat	*MS*, RI, O	**hexanoic acid**		82	82	82	67
30	1880	Green, herb	*MS*, RI, O	3-mercapto-1-hexanol	V	-	33	47	33
31	1835	Tobacco, honey	*MS*, RI, O	2-phenylethyl acetate		47	47	58	33
32	1914	Rose, lilac	*MS*, RI, O	**2-phenylethanol**		82	100	82	82
33	1979	Flower	*MS*, RI, O	β-ionone		47	33	47	47
34	1997	Spice	RI, O	bulnesol		58	47	-	-
35	2038	Caramel, candy	*MS*, RI, O	**4-hydroxy-2,5-dimethyl-3(2*H*)-furanone (furaneol)**	ST	82	67	67	47
36	2079	Cotton candy	*MS*, RI, O	**2-ethyl-4-hydroxy-5-methyl-3(2*H*)-furanone (homofuraneol)**	ST	82	33	33	67
37	2084	Sweat	MS, RI, O	octanoic acid		-	47	47	33
38	2154	Clove, spice	*MS*, RI, O	eugenol	S	67	47	33	33
39	2235	Curry	*MS*, RI, O	**3-hydroxy-4,5-dimethylfuran-2(5*H*)-one (sotolon)**	S	67	47	58	82
40	2353	Sweet, grape must	MS, RI, O	diethyl tartrate		33	67	-	-

^a^ Summarized descriptions of aromas detected at sniffing port by panelists. ^b^ MS (italicized), mass spectrum matches with authentic compound; MS (not italicized), mass spectrum matches with the literature and/or library; RI, retention index matches with literature; O, odor matches with the literature. ^c^ Aroma group codes: F, fruity; V, vegetal; ST, strawberry/sweet; S, spicy/dried/canned fruit; B, berries. ^d^ Undetected.

**Table 5 molecules-28-05016-t005:** Aroma descriptors employed and reference standards for assessor training.

Descriptor	Reference Standard	Amount ^a^
Cherry	standard (Vioryl, Afidnes, Greece, https://www.vioryl.gr/el/)	10 μL
Sour cherry	sour cherry syrup (Jiotis, Athens, Greece, https://www.jotis.gr/)	5 g
Strawberry	standard (Vioryl, https://www.vioryl.gr/el/)	13 μL
Berry fruit	standard (Vioryl, https://www.vioryl.gr/el/)	20 μL
Plum	plum juice (Sunsweet, Yuba City, CA, USA, https://www.sunsweet.gr/)	33 mL
Green bell pepper	green bell pepper (fresh)	8 g 10 cm^2^ pieces
Tomato	tomato paste (Kyknos, Savalia, Greece, https://kyknoscanning.com/el/)	7.5 g
Olive	olives (Edem, Peania, Greece, https://www.edem.com.gr/)	4 g crushed
Mushroom	1-octen-3-ol 1 g/L (Sigma-Aldrich, Saint Louis, MO, USA, https://www.sigmaaldrich.com/)	800 μL
Spices	grated black pepper (Kagia) and grated clove (Kagia, Patra, Greece, https://www.kagiaspices.gr/)	0.1 g and 0.01 g, respectively
Vanilla	vanillin powder (Captain’s, Athens, Greece, http://www.captainspices.gr/)	0.3 g

^a^ Amounts shown were added to 100 mL of deodorized red base wine.

## Data Availability

The data presented in this study are available in the article and Appendix A.

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
