# Peer review of "Chemical and Sensory Characterization of Xinomavro Red Wine Using Grapes from Protected Designations of Northern Greece"

_molecules, 2023, doi:10.3390/molecules28135016_

Round 1

Reviewer 1 Report

Chemical and sensory characterization of Xinomavro red wine using grapes from protected designations of Northern Greece

1.    Please, emphasize the innovativeness of this work. The olfactory profile of this grape variety is well known.

2.    These wines were subjected to a number of tests for the presence of characteristic compounds in order to test their authenticity. How are these studies different? Please specify this in detail in the introduction.

3.    Conclusions should be more related to work performance. Please change it.

4.    Please change some old references. You can replace them with newer ones.

Please describe the innovativeness of this work in comparison to other existing studies.

Reviewer 2 Report

This paper deals with the characterization of a PDO red wine. The orginality is low, however the paper is well written and fluent. In my opinion, the addition of data related with the soil differences and/or agronomical practices would have been useful to comment the data obtained and the differences among the wines. Moreover statistics such as PLSR would have given added value to the paper.

Here you can find some minor revisions

Abstract:

line 28: correct presence in present.

Results and discussion:

Line 97: the figure 1 is redundant because the data were present in the table.

Line 111: add are after "they typically found"

LInes 118-121: this sentence is not clear and must be reformulated.

Line 136: 4.7 is not correct, it should be 3.4 at seeing the table 2.

LIne 154.: add "be" after could...extracted.

Line 171: this is not significant, is higher but not significantly. Add this concept to the text.

Line 176-177: this sentence must be rewritten. Xin4 is similar to all the other.

Line 178: I would put before Xin3, then 4 and 2.

Lines 195-197: It is correlated also with cherry.

Lines 223-225: check this sentence. Rewrite it.

Line 246_250: the same as above.

Lines 256-260: here it would be very useful to add information about soil and agronomical practices.

Lines 279-283: here a PLSR would be very interesting.

Line 336: you write ultimately but you are referring to a paper of 2012.

Line 462: A 1 mL, eliminate A.

Line 490: add the Pearson correlation tha you used.

I woud rewrite completely the conclusions that do not help to emphasize the data obtained.

Table S1 be careful because all the attributes are all significant (p<0.0001) but not in the text.

Round 2

Reviewer 1 Report

After the corrections made, I believe that the article can be published.